# Human adherent cortical organoids in a multi-well format

**Mark van der Kroeg[1], Sakshi Bansal[1], Maurits A Unkel[1], Hilde Smeenk[1], Steven A Kushner[1,2,3]\*†, Femke MS de Vrij[1,4]\*†**

[1]Department of Psychiatry, Erasmus MC, Rotterdam, Netherlands; [2]Stavros Niarchos Foundation (SNF) Center for Precision Psychiatry & Mental Health, Columbia University, New York, United States; [3]Department of Psychiatry, Columbia University Irving Medical Center, New York, United States; [4]ENCORE Expertise Center for Neurodevelopmental Disorders, Erasmus MC, Rotterdam, Netherlands

## eLife Assessment

This paper describes an **important** advance in a 2D in vitro neural culture system to generate mature, functional, diverse, and geometrically consistent cultures, in a 384-well format with defined dimensions and the absence of the necrotic core, which persists for up to 300 days. The well-based format and conserved geometry make it a promising tool for arrayed screening studies. The evidence is **compelling** and provides a method for generating consistent 3D cortical layer-like organization.

**\*For correspondence:**
sk2602@cumc.columbia.edu
(SAK);
f.devrij@erasmusmc.nl (FMSdV)

†These authors contributed
equally to this work

**Competing interest:** The authors
declare that no competing
interests exist.

**Reviewing Editor:** Genevieve
Konopka, David Geffen School of
Medicine at UCLA, United States

**Abstract** In the growing diversity of human induced pluripotent stem cell (iPSC)-derived models of brain development, we present here a novel method that exhibits 3D cortical layer formation in a reproducible topography of minimal dimensions. The resulting adherent cortical organoids (ACOs) develop by self-organization after seeding frontal cortex-patterned iPSC-derived neural progenitor cells in 384-well plates during 8 weeks of differentiation. The organoids have stereotypical dimensions of 3 × 3 × 0.2 mm, contain multiple subtypes of neurons, astrocytes, and oligodendrocyte lineage cells, and are amenable to extended culture for at least 10 months. Longitudinal imaging revealed morphologically mature dendritic spines, axonal myelination, and robust neuronal activity. Moreover, ACOs compare favorably to existing free-floating brain organoid models on the basis of robust reproducibility in obtaining topographically standardized radial cortical structures and circumventing internal necrosis. Adherent human cortical organoids hold considerable potential for high-throughput drug discovery applications, neurotoxicological screening, and mechanistic pathophysiological studies of brain disorders.

## Introduction

Human embryonic or induced pluripotent stem cell (hiPSC)-derived models have yielded considerable success in elucidating neurodevelopmental biology in health and disease (*Levy and Paşca, 2025*). The approaches have been varied and complementary, including single-cell hiPSC-derived models grown in a monolayer (*Sarkar et al., 2018*; *Shan et al., 2024*; *Zhang et al., 2013*), multiple neural cell types in 2D neural networks (*Astick and Vanderhaeghen, 2018*; *Bardy et al., 2015*; *Gunhanlar et al., 2018*; *Shi et al., 2012*), 3D free-floating regionalized neural organoids (*Paşca et al., 2015*; *Qian et al., 2016*; *Ullah et al., 2026*; *Xiang et al., 2019*; *Zhang et al., 2023*), and hiPSC-derived free-floating unguided neural organoids and assembloids (*Gomes et al., 2020*; *Kim et al., 2025*; *Lancaster et al., 2013*; *Pasca et al., 2022*; *Pasca et al., 2025*; *Pellegrini et al., 2020*; *Renner et al.,*

2017; *Sawada et al., 2020*). Although models are continuing to improve, increasing cellular and topographical complexity has appeared to come at the cost of inter-organoid variability (*Eichmüller and Knoblich, 2022*; *Kelava and Lancaster, 2016*). Therefore, a major current technical challenge is to identify hiPSC-derived models that recapitulate higher-order neural complexity with reduced heterogeneity.

Existing 3D models suffer from considerable inter-organoid variability due to the complex and heterogeneous nature of the free-floating structures (*Cederquist et al., 2019*; *He et al., 2024*; *Lancaster et al., 2013*; *Lancaster et al., 2017*; *Paşca et al., 2015*; *Renner et al., 2017*; *Velasco et al., 2019*; *Yoon et al., 2019*). A further challenge, in particular with free-floating organoids, is the resulting necrotic core that frequently occurs when tissue volumes exceed the limits of oxygen and nutrient diffusion beyond a radius of ~300–400 µm (*Lancaster et al., 2017*; *Uzquiano et al., 2022*). Although recent progress has been made with slicing organoids prior to the emergence of necrosis, followed by organotypic air–liquid interface culture (*Giandomenico et al., 2019*; *Qian et al., 2020*), even sliced organoids have to be repeatedly re-cut to prevent necrosis (*Qian et al., 2020*), which is both laborious and risks introducing another potential source of variability. The generation of vascularized organoids is currently a major focus in the field as a potential solution to reducing the necrotic core and some of the inherent cellular stress that is observed in cortical organoids. Moreover, xenotransplantation of organoids and hiPSC-derived neural cells into rodent brains also holds considerable promise as a model system to investigate circuit integration in vivo (*Bhaduri et al., 2020*; *Fan et al., 2022*; *Pasca et al., 2025*; *Schafer et al., 2023*; *Vermaercke et al., 2024*), but relies upon the use of animal models that limits advantages of a fully human cellular system. Increasing diffusion in organoids through microfluidics or vascularization (*Kistemaker et al., 2025*; *Matsui et al., 2021*) would be beneficial. Additionally, further investigation into the mutually supportive interactions between neural and vascular cells in a fully in vitro system composed exclusively of human cells is warranted (*Crouch et al., 2022*; *Mansour et al., 2018*; *Wang et al., 2023*).

Here, we propose a simplified approach to generating long-term hiPSC-derived adherent cortical organoids (ACOs) with reproducible dimensions below metabolic diffusion limits in a readily screenable 384-well format. ACOs can be maintained in long-term culture and contain neurons with dendritic spines and robust activity, as well as several classes of glial cells including oligodendrocyte precursor cells (OPCs), myelinating oligodendrocytes, and morphologically distinct subtypes of astrocytes.

## Results

### Self-organized topography of iPSC-derived ACOs

ACOs reproducibly self-organize into layered radial structures in 384-well plates within 8 weeks of seeding with hiPSC-derived purified forebrain-patterned neural progenitor cells (NPCs). Three different hiPSC source cell lines were used in this study, including commercially available NPCs, and NPCs generated using a modified version of our previously described protocol (*Gunhanlar et al., 2018*; *Figure 1A*). NPCs were capable of neural rosette formation and expressed SOX2, Nestin, and the frontal cortical NPC-marker FOXG1 (*Figure 1B, D*, *Figure 1—figure supplement 1A*). The initial 4 weeks after seeding the NPCs in 384-well plates in Neural Differentiation Medium (ND) were characterized by proliferative expansion of NPCs and the emergence of early neural differentiation markers (*Figure 1E*, *Figure 2—figure supplement 1*). Between 4- and 8-week post-seeding, neurons and glial cells emerged with a consistent spatial organization (*Figure 1E, F*; *Figure 1—figure supplement 1B*, *Figure 2—figure supplement 1*), in which the central region was densely packed with cell bodies while the periphery contained circumferentially and radially organized processes originating from cells in the center. Typically, ACOs with a single structure were observed in ~80% of the wells seeded with NPCs after 60 days of differentiation. While some later batches show moderately reduced success rates compared with the earliest batches, properly formed single structure organoids were still obtained at 40–90% success across all examined time points (*Figure 1—figure supplement 1C*), indicating that long-term culture is feasible albeit with variable efficiency. Organoid topography was highly dependent on the proliferation rate of NPCs, which can differ substantially across differentiation batches and hiPSC clones. For each NPC line, optimal seeding density was estimated based on the proliferation rate of that NPC line. Multiple densities were seeded around the estimated optimal density, and after 6–8 weeks, the NPC density at which ACO generation was most reliably generated was determined

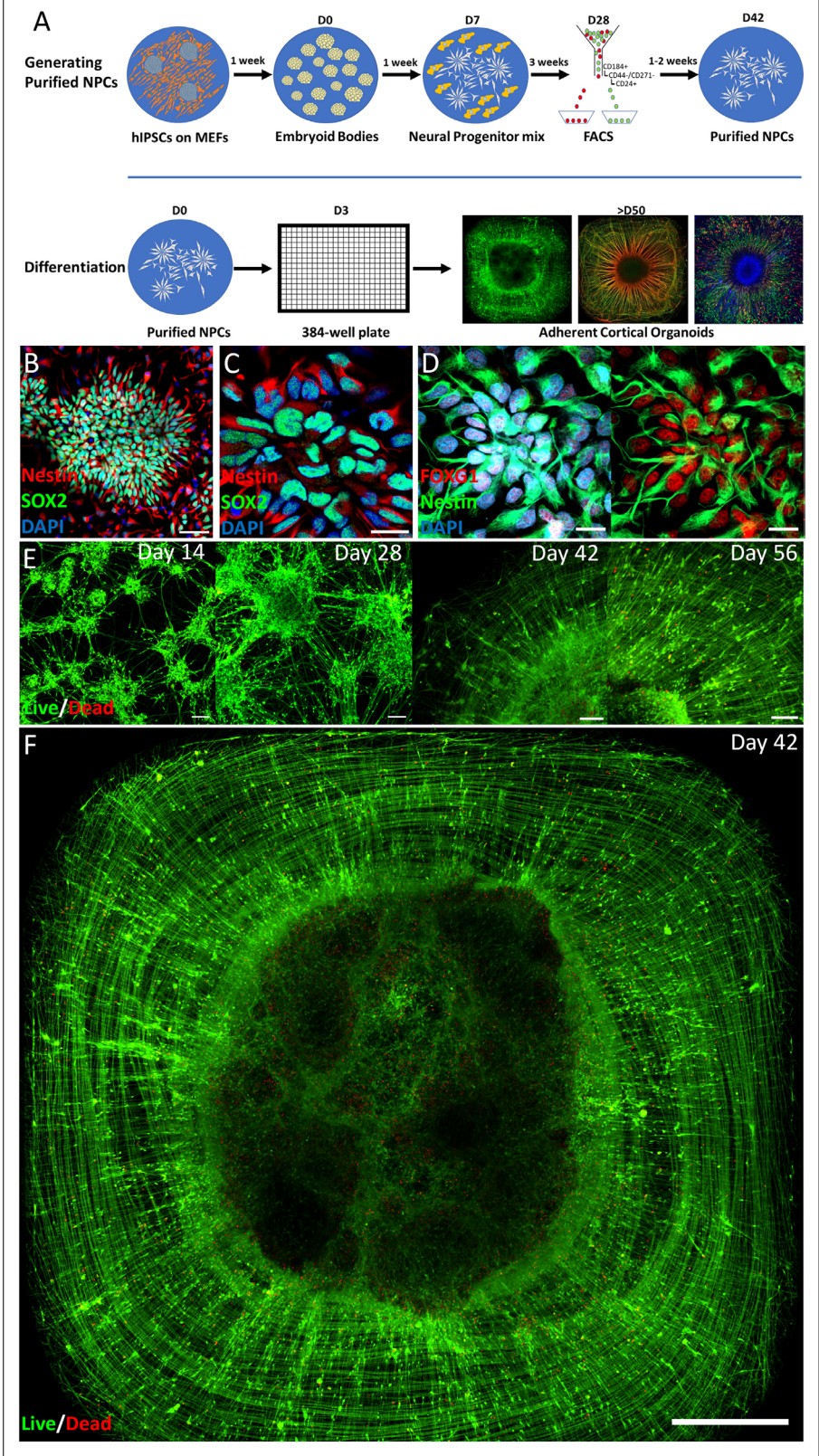

**Figure 1.** Adherent cortical organoid model. (**A**) Schematic representation of the differentiation protocol. (**B–D**) Representative pictures of neural progenitor cells (NPCs) from three different induced pluripotent stem cell (iPSC) lines with markers SOX2, Nestin, and FOXG1 (scale bar B, 50 µm; C, D, 20 µm). (**E**) Live/Dead stain of a representative time course showing self-organization during differentiation, starting with radial organization

*Figure 1 continued on next page*

*Figure 1 continued*

between days 28 and 42, seeding density 1500 NPCs per well; scale bars left to right 100, 150, 100, and 100 μm. (**F**) Full well showing radial organization at day 42 in culture were only a few dead cells are visible in red in the dense center of the structure, seeding density 1500 NPCs per well (scale bar, 500 μm).

The online version of this article includes the following figure supplement(s) for figure 1:

**Figure supplement 1.** Reproducibility of neural progenitor cells (NPCs) and adherent cortical organoids.

empirically using a combination of visual inspection, Live/Dead staining, and immunocytochemistry. The proliferation rate and seeding density of NPCs required for generating topographically reliable ACOs were correlated ($r^2$ = 0.67) and provide guidance for empirically determining the optimal NPC seeding density for a given line (*Figure 1—figure supplement 1D*).

## Cell type distribution and layer formation

Tau+/MAP2− axons exhibited long extensions in a circular pattern, while MAP2+ dendrites exhibited orthogonally oriented radial outgrowth (*Figure 2A, B*, *Figure 2—figure supplement 2*). The spatial organization that evolved over the first 8 weeks after seeding was paralleled by a shift in cell type distribution.

Overall, a reduction in progenitor markers (SOX2 day 14: 58.4%, day 56: 18.8%, p ≤ 0.001; PAX6 day 14: 34.5%, day 56: 8.0%, p = 0.20) and a significant increase in neuronal cortical layer markers (CTIP2 day 14: 0.5%, day 56: 14.0%, p ≤ 0.001; CUX1 day 14: 1.3%, day 56: 24.4%, p ≤ 0.001) were observed (*Figure 2—figure supplement 1*). However, a SOX2-positive layer remained present in the center of the ACOs at day 63 (*Figure 2—figure supplement 3C*). Cortical layer markers exhibited an inside-out pattern of development in which expression of the deep-layer excitatory neuronal marker CTIP2 emerged before the upper-layer marker CUX1 (*Figure 2C, E*, *Figure 2—figure supplement 1*). Six to eight weeks following seeding, a self-organized rudimentary segregation of deep- and upper-layer neurons emerged as shown by the macroscopic separation of deep- and upper-layer neurons, although some neurons were spatially intermixed and some neurons were double-positive for CTIP2 and CUX1. In the upper-layer neuronal region, segregation was observed between CUX1- and CUX2-positive cells (*Figure 2E*, *Figure 2—figure supplement 3*) as CUX2 is typically expressed over a wider range of upper cortical layers than CUX1 and also marks intermediate progenitors (*Molyneaux et al., 2007*). Analogous to the broad distribution of cortical cell subclasses, the majority of the neurons were glutamatergic, while GAD67+ interneurons were also present (*Figure 2F*), constituting ~10% of the NeuN+ neuronal population consistently for all three source hiPSC lines (*Figure 2G*). Over a developmental time course, GAD67/NeuN remained relatively stable, although it showed a slight increase at differentiation day 114 (*Figure 2H*).

## ACOs contain multiple glial cell types

Within 8 weeks of seeding, a population of GFAP+/S100β+ astrocytes emerged. Many astrocytes had their soma located in the central region with process outgrowth radially (*Figure 2I*, *Figure 2—figure supplement 4*), while other astrocytes exhibited subtype-specific morphologies including fibrous astrocytes (*Figure 2J*), protoplasmic-like astrocytes (*Figure 2K*), and interlaminar astrocytes (*Figure 2L*). GFAP/PAX6 double-positive radial glia were present at the outskirts of the densely populated center of the organoid (*Figure 2M*).

Similar to free-floating organoids, ACOs survived for longer periods compared to monolayer neural cultures grown on larger surfaces. This longevity allows for the development of cell types not usually seen in a monolayer culture that can typically be cultured up to a maximum of 2–3 months. By 6 weeks after seeding NPCs, we observed the emergence of OPCs, as shown by the expression of NG2 (*Figure 3A*), and the NG2+ cell population remained present when the first oligodendrocytes emerged (*Figure 3B, C*). Staining for Myelin Basic Protein (MBP) revealed the emergence of MBP+ oligodendrocytes around 4 months after NPC seeding when the ACOs were continuously grown in the presence of T3 (2 ng/ml) (*Figure 3D*). At 5 months, the oligodendrocytes showed increasingly mature morphologies (*Figure 3E, I*) and exhibited MBP co-localization along NF200+ axons (*Figure 3F, I*).

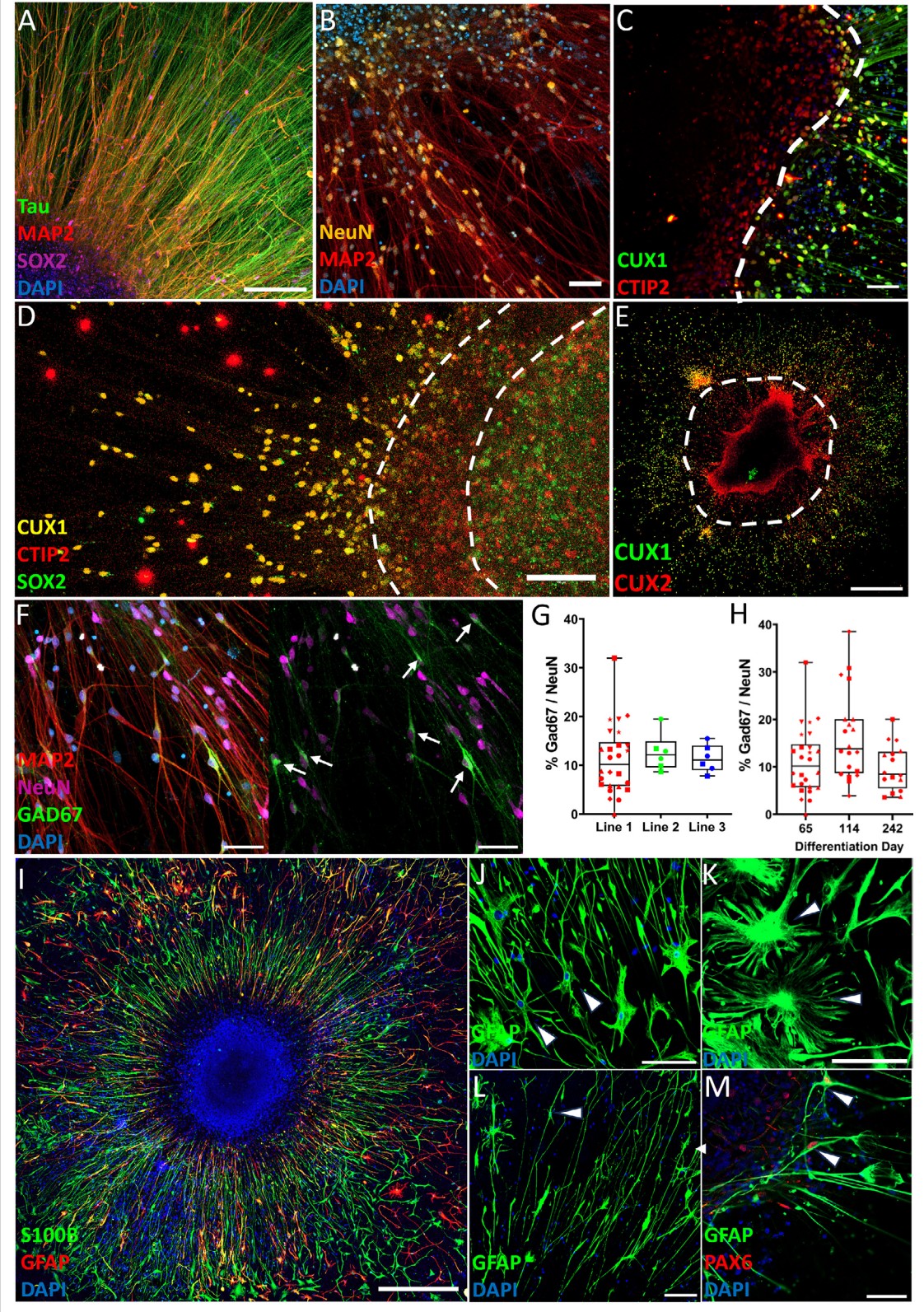

**Figure 2.** Adherent cortical organoids show an organized network of neuronal and astrocyte subtypes. (**A**) MAP2+ somas and dendrites alongside Tau+/MAP2– axons show segregation of dendritic and axonal compartments, with SOX2+ progenitors concentrated in the center of the organoid (day 75, 200 µm). (**B**) MAP2+ and NeuN+ cells indicate mature neurons (day 72, 50 µm). (**C**) Deep-layer cortical marker CTIP2 and upper-layer marker CUX1 show rudimentary segregation of cortical layers in expected inside-out pattern (day 64, 50 µm). (**D**) Rudimentary separation of SOX2+ neural progenitor

*Figure 2 continued*

cells (NPCs), CTIP2+ deep-layer neurons, and CUX1+ upper-layer neurons (day 67, scale bar, 100 μm). (**E**) Full well overview of the CUX1- and CUX2-positive regions. (**F**) GAD67+ interneuron (white arrows) proportion of NeuN+ neurons (day 67, 50 μm). (**G**) Quantification of GAD67+ proportion of NeuN+ neurons. Every point is the percentage of GAD67+ interneurons out of all NeuN+ neurons in a spatially randomized selected image with on average 130.6 (±16.1) NeuN+ nuclei per frame (line 1: *N* = 26 images of 6 different cortical organoids at days 65–67; line 2/3: 6 images of 2 different organoids at day 65). (**H**) Quantification of GAD67+ proportion of NeuN+ neurons of cell line 1, at three different time points. The number of GAD67+ interneurons over the total NeuN+ population remains relatively stable although at day 114 there was a modest increase which disappeared at day 242. Data was collected from 4 to 6 organoids per time point. (**G, H**) The different symbols reflect from which organoid the data point was collected. (**I**) Astrocyte markers GFAP and S100β show the general radial pattern of astrocyte outgrowth (day 66, 500 μm). GFAP staining reveals the morphologies of different astrocyte subtypes (white arrows), including fibrous astrocytes (**J**, day 65, 100 μm), protoplasmic astrocytes (**K**, day 65, 50 μm), and interlaminar astrocytes (**L**, day 65, 100 μm). (**M**) Co-localization of astrocyte marker GFAP and PAX6 marks radial glia (white arrows) (day 65, 50 μm).

The online version of this article includes the following figure supplement(s) for figure 2:

**Figure supplement 1.** Neuronal maturation in cortical organoids is shown by a temporal shift in neural progenitor cell (NPC) and neuronal markers.

**Figure supplement 2.** Distribution of axons and dendrites in adherent cortical organoids Radially organized MAP2+/Tau+ dendrites along with both radially and circumferentially organized NF200+/Tau+ axons in duplicate for each cell line (day 63, scale bars, 500 μm).

**Figure supplement 3.** Cortical layering in the adherent cortical organoids (ACOs).

**Figure supplement 4.** Astrocyte distribution within adherent cortical organoids (ACOs).

## ACOs show synaptic connectivity and functional activity

Neurons within ACOs exhibited robust synaptogenesis (*Figure 4A, E*). Sparse labeling of excitatory neurons with AAV9.CamKII.eGFP revealed the presence of Synapsin-positive (Syn+) mushroom-shaped dendritic spines (*Figure 4E*). 51.5% of the Syn+ pre-synaptic boutons co-localized with the excitatory post-synaptic density marker HOMER1 (*Figure 4F, I*), with a density of 10.9 Synapsin/HOMER1 double-positive synapses per 100 μm$^2$ MAP2 (*Figure 4H*). Moreover, 21.1% of Syn+ pre-synaptic boutons co-localized with the inhibitory post-synaptic density marker Gephyrin (*Figure 4G–I*), with a density of 5.0 Synapsin/Gephyrin double-positive synapses per 100 μm$^2$ MAP2. At differentiation day 114, we observed a ~2:1 ratio of excitatory to inhibitory synaptic contacts (*Figure 4H*).

To assess the functional activity of the ACOs, we used the genetically encoded calcium indicator GCaMP6s under the control of the human Synapsin promoter (*Figure 5A*). Calcium imaging revealed robust synchronous network-level bursting (NB) (1.4 ± 0.07 NB/min) at day 61, which remained stable until at least day 100 (1.6 ± 0.12 NB/min), in which all recorded neurons participated to varying degrees (*Figure 5B, D*). In addition, substantial desynchronized activity was also observed outside of network-level bursting while the total activity remained stable over time (day 61, 3.9 ± 0.5 events/min; day 100, 3.9 ± 0.6 events/min) (*Figure 5B, E*, *Figure 5—video 1*).

## Discussion

The study of early human brain development and related diseases has long been hampered by the inherent complexity of the human brain and the inaccessibility of living brain tissue at cellular resolution. Technological advances in iPSC technology have now facilitated the opportunity to obtain living human neurons derived from specific individuals.

Here, we describe a platform to model early human frontal cortical development with high reproducibility and simplified organization. While 3D free-floating organoids and sliced organoids recapitulate layered cortex formation, they are subject to variation in the relative contribution of cortical tissue within the organoid, forming multiple cortical patches along the edges and complicating structured analysis (*Eichmüller and Knoblich, 2022*; *Giandomenico et al., 2019*; *Quadrato et al., 2017*). Moreover, 3D free-floating organoids suffer from necrosis in the core of the organoid due to lack of oxygen and nutrient diffusion. Recently, other protocols have been published starting from rosette formation with a focus on early neurodevelopment and leading to single structure free-floating cortical organoids (*Pagliaro et al., 2023*; *Tidball et al., 2023*). Our platform predefines a rosette-forming iPSC-derived cortical NPC population that self-organizes into adherent singular radial structures in a standard 384-well format. Other examples have highlighted the benefits of a multi-well format or systematic individual structure formation (*Knight et al., 2018*; *Medda et al., 2016*). Our platform now integrates these features to yield individual adherent layered cortical structures with synaptic connectivity and neuronal activity, as well as the major neural progenitor-derived glial cell types including

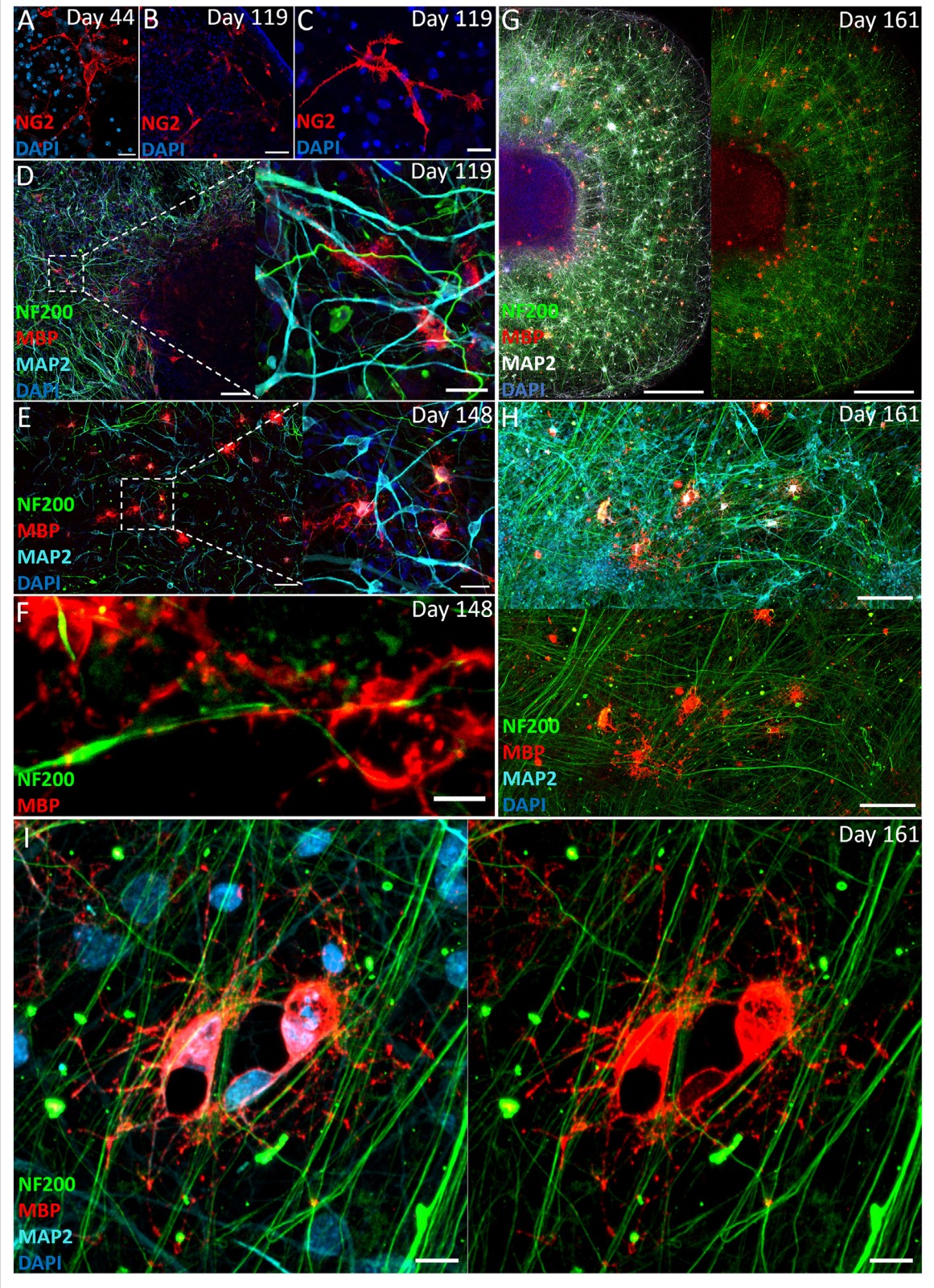

**Figure 3.** Adherent cortical organoids form oligodendrocyte lineage cells. (**A**) Adherent cortical organoids show oligodendrocyte precursor cells (OPCs) as early as 44 days, indicated by OPC marker NG2 (day 44, 20 μm). The NG2+ OPCs are still present in the cortical organoids after 4 months (day 119, **B** 50 μm, **C** 20 μm). (**D**) Young oligodendrocytes start to emerge after 4 months indicated by rudimentary MBP staining (day 119, 100 and 20 μm). After 5 months, MBP-positive oligodendrocytes show more mature morphology and initial wrapping of NF200+ axons (day 148; **E**, 50 and 20 μm, **F**, 5 μm).

*Figure 3 continued on next page*

*Figure 3 continued*
Oligodendrocyte distribution at day 161 where the MBP+ oligodendrocytes sit between axon bundles and co-localize with NF200+ axons (**G–I**, day 161, G 500 µm, H 100 µm, I 10 µm).

OPCs, myelinating oligodendrocytes, and astrocytes. The reproducible format of adherent organoids in a 384-well format has the distinct advantage of being able to image entire organoids without slicing or clearing, including under live cell conditions. We confirmed the self-organizing potential and reproducibility of ACOs across multiple hiPSC lines and using different sources of NPCs, controlling the seeding density for the proliferation rate of the specific NPC batch. Seeding NPCs with frontal cortical identity in the defined geometry of a 384-well plate enabled the development of long-term functional neural networks in a complex radial structure resembling early human cortical development. Future studies aimed at single-cell RNA sequencing and advanced image-based analysis solutions in this platform will be important to further benchmark adherent organoids with topographical, typological, and temporal hierarchies in the developing human cortex (*Bhaduri et al., 2021*; *Nowakowski et al., 2017*; *Nowakowski et al., 2016*; *Uzquiano et al., 2022*). Functional ACOs in a multi-well format have the potential to be leveraged for high-throughput screening applications, neurotoxicological studies, mechanistic pathophysiological studies of neurodevelopmental and neuropsychiatric disorders, and pharmacological and phenotypic screening of disease phenotypes during early cortical development.

ACOs also have some limitations, most notably that cortical layering and regional specification appear to be more advanced in free-floating brain organoids, including gyrification in some models. Key characteristics of the model, including concurrent neuronal and astrocyte differentiation and a consistent ACO structure, were robust across hiPSC lines. Because oligodendrocytes arise later and require longer culture times, their generation showed greater variability. We are currently optimizing this aspect of the ACO model. Therefore, the choice of model system should be made based on the specific experimental question being addressed.

Taken together, we present a novel platform for cellular-level human brain modeling using ACOs that exhibit high reproducibility and robust neuronal activity. The ability to reliably generate human cortical organoids in multi-well plates combined with neural network functionality offers a unique potential for brain disease modeling and therapeutic screening applications.

## Materials and methods
### Generation of NPCs

NPCs from three different source hiPSC lines were used. NPC-line 1: in-house generated NPCs from human iPSC line WTC11 (provided by Bruce R. Conklin, The Gladstone Institutes and UCSF, #GM25256, RRID:CVCL_Y803, *Miyaoka et al., 2014*). NPC-line 2: commercially available hNPCs from Axol Biosciences (ax0015). NPC-line 3: NPCs derived using the protocol of *Shi et al., 2012* from hiPSC line IPSC0028 (Sigma-Aldrich, RRID:CVCL_EE38). Line 1 NPCs were generated from the WTC11 hiPSC line grown on mouse embryonic fibroblasts (MEFs) in human embryonic stem cell medium (Dulbecco's modified Eagle's medium (DMEM)/F12 (Thermo Fisher Scientific)), 20% knockout serum (Thermo Fisher Scientific), 1% minimum essential medium/non-essential amino acid (Sigma-Aldrich, St Louis, MO, USA), 7 nl ml β-mercaptoethanol (Sigma-Aldrich), 1% L-glutamine (Thermo Fisher Scientific), and 1% penicillin/streptomycin (Thermo Fisher Scientific). Mycoplasma contamination was routinely tested using the MycoAlert bioluminescence-based assay (Lonza), and all cell cultures were consistently negative. The hiPSC colonies were dissociated from the MEFs with collagenase (100 U ml, Thermo Fisher Scientific, Waltham, MA, USA) for 7 min at 37 °C/5% $CO_2$. Embryoid bodies (EBs) were generated by transferring dissociated iPSCs to non-adherent plates in human embryonic stem cell medium on a shaker at 37°C/5% $CO_2$. EBs were grown for 2 days in human embryonic stem cell medium, changed into neural induction medium (Advanced DMEM/F12, 1% N2 supplement (Thermo Fisher Scientific), 2 µg ml heparin (Sigma-Aldrich), and 1% penicillin/streptomycin) on day 2 (d2) and cultured for another 5 days in suspension (d3–d7). The EBs were slightly dissociated at d7 by trituration and plated onto laminin-coated 10 cm dishes (20 µg ml laminin (Sigma-Aldrich) in DMEM for 30 min at 37°C), initially using neural induction medium (d7–d14), and then from d15 in NPC medium consisting of: Advanced DMEM/F12 (Life Technologies), 1% N2 supplement (Thermo Fisher Scientific), 2% B27-RA supplement (Thermo Fisher Scientific), 1 µg/ml laminin (L2020, Sigma-Aldrich), 20 ng/ml

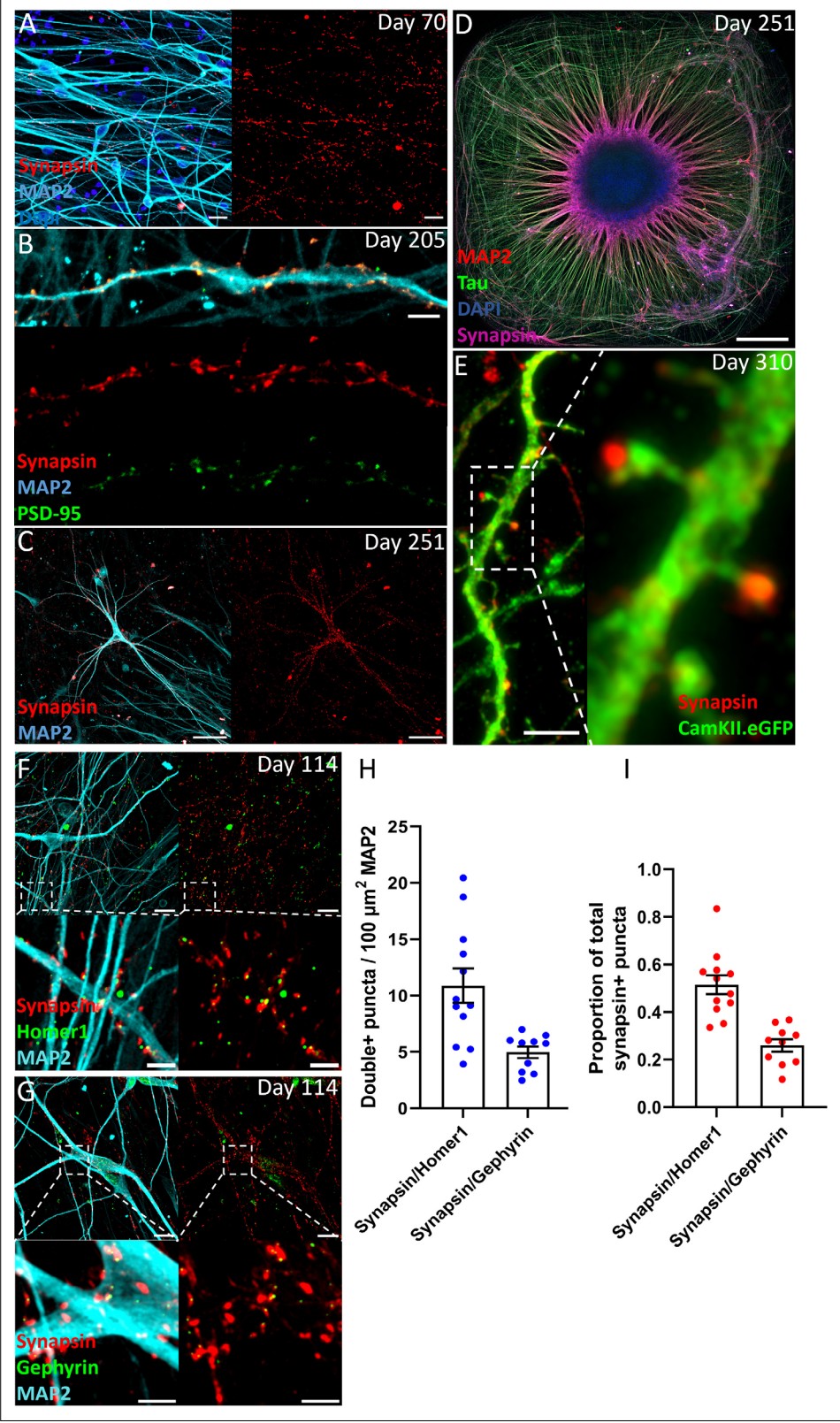

**Figure 4.** Adherent cortical organoids show a neural network with excitatory and inhibitory synapses. (**A**) Synapsin staining shows synapse formation along MAP2+ dendrites (day 70, 20 µm). (**B**) Co-localization of pre-synaptic marker Synapsin and post-synaptic marker PSD-95 (day 205, 20 µm). (**C**) MAP2+ dendrites and soma are decorated with Synapsin+ synapses (day 251, 100 µm). (**D**) Overview of entire well, showing alignment of Synapsin staining

*Figure 4 continued on next page*

*Figure 4 continued*

with MAP2 (day 251, 500 µm). (**E**) Sparse labeling of neurons with AAV9.CamKII.eGFP allows detailed imaging of glutamatergic dendritic spines with mature mushroom morphology, contacting pre-synaptic Synapsin puncta (day 310, 5 µm). (**F**) Excitatory synapses marked by overlapping Synapsin+ and HOMER1+ puncta (scale bar: top 20 µm, bottom 5 µm). (**G**) Inhibitory synapses marked by overlapping Synapsin+ and Gephyrin+ puncta (scale bar: top 20 µm, bottom 5 µm). (**H**) Density of excitatory and inhibitory synapses per 100 µm². There are about twice as many excitatory synapses (10.9/100 µm²) as inhibitory synapses (5.0/100 µm²). (**I**) The proportion of Synapsin that co-localizes with HOMER1 is 0.51 and the proportion of Synapsin that co-localizes with Gephyrin is 0.21.

basic fibroblast growth factor (Merck-Millipore, Darmstadt, Germany), and 1% penicillin/streptomycin (Thermo Fisher Scientific). After passage 3, NPC cultures were purified using fluorescence-activated cell sorting (FACS). NPCs were detached from the culture plate using Accutase (Stem Cell Technologies) and CD184+/CD44−/CD271−/CD24+ cells (*Yuan et al., 2011*) were collected on a FACSAria III Cell Sorter (BD Bioscience) and expanded in NPC medium. NPCs were maintained in NPC medium on 6-well plates (Sanbio, 140675) coated with 20 µg/ml laminin (Sigma, L2020) in Advanced DMEM/F12 (Life Technologies) and were used for differentiation to ACOs between passages 3 and 7 after the FACS purification step.

## Generating ACOs

NPCs were dissociated with Accutase (Stem Cell Technologies), and live cells were counted with Trypan Blue (Stem Cell Technologies) in a Burker counting chamber. The NPCs were seeded in 384-well plates (M1937-32EA. Life Technologies) that were coated with 50 µg/ml laminin in Neurobasal medium (Thermo Fisher Scientific) for at least 30 min at 37°C. The required seeding cell densities to generate ACOs for each individual cell line were determined by seeding a range of cell densities followed by visual inspection with a bright field microscope, Live/Dead staining, and immunocytochemistry. When too few NPCs are seeded, the organoids will not form and show patches of clustered cells like you would see in a standard monolayer culture. When too many NPCs are seeded, the 384-plate well will overgrow, leading to a lack of organoid structure formation. The variation in NPC seeding density is mostly explained by NPC proliferation rate, and to a lesser degree by the capacity of a given cell line to differentiate into post-mitotic neural cells. NPCs were seeded and differentiated in Neural Differentiation Medium: Neurobasal medium (Thermo Fisher Scientific), 1% N2 supplement (Thermo Fisher Scientific), 2% B27-RA supplement (Thermo Fisher Scientific), 1% minimum essential medium/non-essential amino acid (Stem Cell Technologies), 20 ng/ml brain-derived neurotrophic factor (ProSpec Bio), 20 ng/ml glial cell-derived neurotrophic factor (ProSpec Bio), 1 µM dibutyryl cyclic adenosine monophosphate (Sigma-Aldrich), 200 µM ascorbic acid (Sigma-Aldrich), 2 µg/ml laminin (Sigma-Aldrich), and 1% penicillin/streptomycin (Thermo Fisher Scientific). For oligodendrocyte maturation, the cells were continuously grown in the presence of 2 ng/ml T3 (Sigma-Aldrich). Every 2–3 days the ACOs were refreshed with a 75% media change.

## Immunocytochemistry

For live–dead staining, living cultures were incubated with LIVE/DEAD Viability/Cytotoxicity Kit according to the manufacturer's instructions (Thermo Fisher Scientific). For immunocytochemistry, ACOs were fixed for 20–30 min using 4% formaldehyde in phosphate-buffered saline (PBS), washed with PBS three times, and blocked for 1 hr by pre-incubation in staining buffer containing 0.05 M Tris, 0.9% NaCl, 0.25% gelatin, and 0.5% Triton X-100 (pH 7.4). Primary antibodies were incubated for 48–72 hr at 4°C in staining buffer, washed with PBS, and incubated with the secondary antibodies in staining buffer for 2 hr at room temperature. The cultures were embedded in Mowiol 4-88 (Sigma-Aldrich), after which confocal imaging was performed with a Zeiss LSM700 and Zeiss LSM800 confocal microscope using ZEN software (Zeiss, Oberkochen, Germany). The following primary antibodies were used: SOX2 (Merck-Millipore AB5603, 1:200); Nestin (Merck-Millipore MAB5326, 1:200); MAP2 (Synaptic Systems 188004, 1:200); NeuN (Merck ABN78, 1:200); GFAP (Merck-Millipore AB5804, 1:300); FOXG1 (Abcam AB18259, 1:200); CUX1 (Abcam AB54583, 1:200); CTIP2 (Abcam AB18465, 1:100); Synapsin 1/2 (Synaptic Systems 106003, 1:200); PSD95 (Thermo Fisher Scientific MA1-046, 1:100); HOMER1 (Synaptic Systems 160011, 1:100); Gephyrin (Synaptic Systems 147011, 1:100); Tau (Cell Signaling Technology 4019, 1:200); S100β (Sigma-Aldrich S2532, 1:200); Pax6 (Santa Cruz

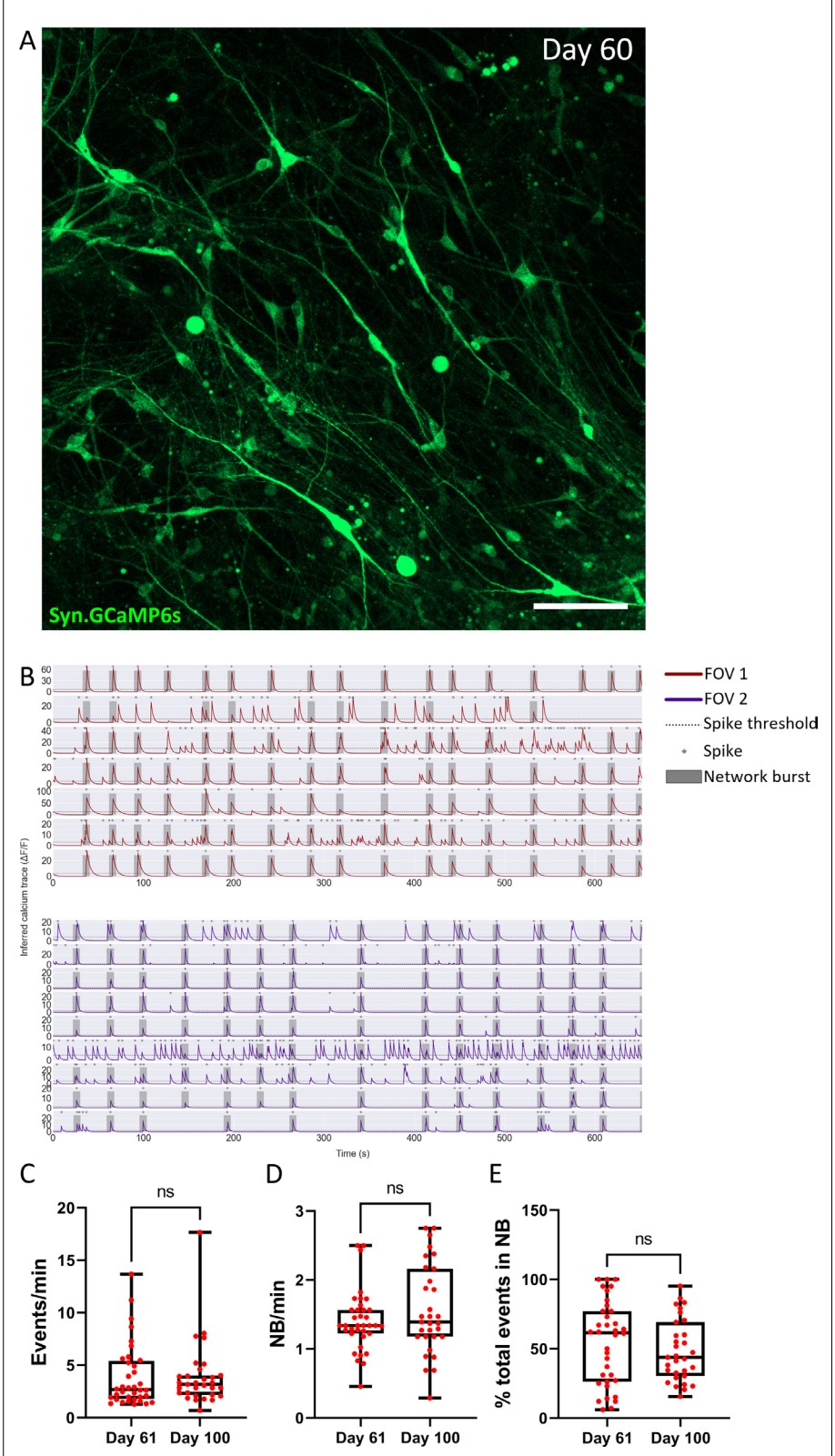

**Figure 5.** Activity in the adherent cortical organoids. (**A**) Snapshot of radially organized neurons transduced with AAV1.Syn.GCaMP6s.WPRE.SV40 (day 60, 100 µm). (**B**) Representative calcium traces from active neurons in two different fields of view (FOVs) from cell line 1, showing both individual activity and network bursts where each line corresponds to a single cell. (**C**) Events per minute of all recorded cells in days 61 and 100 (day 61, 3.9 ± 0.5

*Figure 5 continued on next page*

*Figure 5 continued*

events/min; day 100, 3.9 ± 0.6 events/min). (**D**) Network bursts per minute (D61, average of 1.4 ± 0.07 NB/min; D100 average of 1.6 ± 0.12 NB/min), which shows all active cells are participating with the network bursts to some degree. (**E**) Percentage of events that are part of network bursts, which indicates that some cells only have events that are part of a network burst while for other cells the network bursts are only a small part of their total calcium events. The data was collected from 2 to 3 organoids per time point from 2 different batches of differentiation.

The online version of this article includes the following video for figure 5:

**Figure 5—video 1.** Video showing 10 min of calcium imaging of an adherent cortical organoid (ACO) at day 61 using the genetically encoded calcium indicator AAV1.Syn.GCaMP6s.

https://elifesciences.org/articles/98340/figures#fig5video1

---

sc-81649, 1:100); NG2 (Gift from W. Stallcup Lab, 1:100); NF200 (Sigma-Aldrich 083M4833, 1:200); MBP (Abcam AB7349, 1:100); GFP (Abcam ab13970, 1:100); GAD67 (Merck-Millipore MAB5406, 1:100); DAPI (Thermo Fisher Scientific D1306). The following secondary antibodies were used 1:200: Alexa-488, Alexa-555, and Alexa-647 (Jackson ImmunoResearch, West Grove, PA, USA).

## Sparse labeling of excitatory neurons

pENN.AAV9.CamKII.4.eGFP.WPRE.rBG (Addgene viral prep # 105541-AAV9) was added to the cortical organoids at day 278 ($1.68 \times 10^8$ GC/well). The transduced cortical organoids were fixed and stained on day 310.

## Synaptic quantification

Following immunocytochemistry, confocal imaging was performed with a Zeiss LSM800 confocal microscope with Airy scan using ZEN software (Zeiss, Oberkochen, Germany). Multiple images per organoid were taken of 157.94 µm by 157.94 µm with a depth of 10–15 µm with a scan every 0.17 µm. The surface area of MAP2 and the number of Synapsin+ pre-synaptic terminals quantification was performed on a z-projection of the images in Fiji (*Schindelin et al., 2012*). The co-localization of Synapsin with either HOMER1 or Gephyrin was manually counted. The day 114 data in *Figure 4H, I* was obtained from 4 organoids each with a total area of ~50,000 µm² MAP2 and ~10,000 Synapsin+ pre-synaptic terminals analyzed.

## Calcium imaging

For calcium imaging, the genetically encoded calcium indicator AAV1.Syn.GCaMP6s.WPRE.SV40 (Penn Vector Core, 100843-AAV1) was added to the organoids at day 42 of differentiation ($1.5 \times 10^8$ GC/well). Recordings were performed at day 60 on a Zeiss LSM800 confocal microscope using ZEN software (Zeiss, Oberkochen, Germany). The recordings were made with a 20x/0.8NA Ph2 Plan-Apochromat objective, with a field of view of 150 × 100 µm and a pixel size of 0.3 µm. The acquisition rates of the recordings were between 4 and 5 f.p.s. Twenty-four hours before the recordings, the medium was switched to BrainPhys Neural Differentiation Medium. BrainPhys Neuronal Media (Stem Cell Technologies), 1% N2 supplement (Thermo Fisher Scientific), 2% B27-RA supplement (Thermo Fisher Scientific), 1% minimum essential medium/non-essential amino acid (Stem Cell Technologies), 1% penicillin/streptomycin (Thermo Fisher Scientific), 20 ng/ml brain-derived neurotrophic factor (ProSpec Bio), 20 ng/ml glial cell-derived neurotrophic factor (ProSpec Bio), 1 µM dibutyryl cyclic adenosine monophosphate (Sigma-Aldrich), 200 µM ascorbic acid (Sigma-Aldrich), and 2 µg/ml laminin (Sigma-Aldrich). The calcium imaging recordings were processed using CNMF-E (*Pnevmatikakis et al., 2016*) and FluoroSNAPP (*Patel et al., 2015*) to generate the traces. Calcium traces were then analyzed using a custom script for event and network burst detection using an algorithm written in Python (v3.8.2) (code available at https://github.com/deVrijLab/calcium_imaging copy archived at *Unkel, 2026*). A network burst is defined here as synchronous calcium events of at least 60% of the active neurons in the field of view.

## Cyquant proliferation assay

CyQUANT Direct Cell Proliferation Assay, C35011 (Thermo Fisher Scientific) was used according to the manufacturer's specifications. For each time point and each of the 3 NPC lines, 10–12 wells of a 96-well plate were seeded with NPCs (2500 NPCs per well). In addition, NPC lines generated from

two different clones from IPS line MH0159020 (Rutgers University Cell and DNA Repository) were added to increase the dynamic range of the proliferation curve. At 24 and 96 hr NPCs were frozen at −80°C. All NPC lines and time points were thawed, lysed, and measured together. Doubling time was calculated between 24 and 96 hr. None of the wells were confluent at 96 hr.

## Statistical analysis

All data represent mean ± SEM. When comparing developmental markers in Figure S2, we used one-way ANOVA followed by Tukey–Kramer's multiple correction test. $n$ = 3–6 images taken over two wells, for each time point. To compare the time points in *Figure 5C–E* we used a Mann–Whitney test.

## Acknowledgements

This work was supported by the Netherlands Organ-on-Chip Initiative, an NWO Gravitation project funded by the Ministry of Education, Culture and Science of the government of the Netherlands (024.003.001) to SAK, Hersenstichting Fellowship (F2012(1)-39) to FMSdV, an Erasmus MC Human Disease Model Award to FMSdV, and a Dutch ZonMw MKMD Create2Solve grant (114025201) to SAK and FMSdV.

## Additional information

### Funding

| Funder | Grant reference number | Author |
|---|---|---|
| Nederlandse Organisatie voor Wetenschappelijk Onderzoek | 024.003.001 | Maurits A Unkel<br>Steven A Kushner<br>Femke MS de Vrij |
| Hersenstichting | (F2012(1)-39 | Femke MS de Vrij |
| ZonMw | 114025201 | Mark van der Kroeg<br>Sakshi Bansal<br>Steven A Kushner<br>Femke MS de Vrij |

The funders had no role in study design, data collection, and interpretation, or the decision to submit the work for publication.

### Author contributions

Mark van der Kroeg, Conceptualization, Formal analysis, Validation, Investigation, Visualization, Methodology, Writing – original draft, Writing – review and editing; Sakshi Bansal, Validation, Investigation, Visualization, Methodology; Maurits A Unkel, Formal analysis, Methodology; Hilde Smeenk, Investigation; Steven A Kushner, Conceptualization, Supervision, Funding acquisition, Writing – original draft, Writing – review and editing; Femke MS de Vrij, Conceptualization, Resources, Formal analysis, Supervision, Funding acquisition, Investigation, Methodology, Writing – original draft, Project administration, Writing – review and editing

### Author ORCIDs

Mark van der Kroeg ⬡ https://orcid.org/0000-0001-5517-3687
Sakshi Bansal ⬡ https://orcid.org/0009-0005-0062-888X
Maurits A Unkel ⬡ https://orcid.org/0000-0003-1920-6001
Hilde Smeenk ⬡ https://orcid.org/0000-0002-5529-4857
Steven A Kushner ⬡ https://orcid.org/0000-0002-9777-3338
Femke MS de Vrij ⬡ https://orcid.org/0000-0003-0825-3806

Reviewer #3 (Public review): https://doi.org/10.7554/eLife.98340.3.sa1
Author response https://doi.org/10.7554/eLife.98340.3.sa2

## Additional files

### Supplementary files
MDAR checklist

### Data availability
Raw calcium imaging videos, ROIs, tracing, and analysis are deposited as an openly available dataset on DataverseNL: https://doi.org/10.34894/5E8AHT. The code for the calcium imaging analyses presented in this paper is openly accessible at https://github.com/deVrijLab/calcium_imaging (copy archived at *Unkel, 2026*).

The following dataset was generated:

| Author(s) | Year | Dataset title | Dataset URL | Database and Identifier |
|---|---|---|---|---|
| van der Kroeg M, Bansal S, Unkel M, Kushner SA, de Vrij FMS | 2026 | Calcium imaging adherent cortical organoids Van der Kroeg et al 2026 | https://doi.org/10.34894/5E8AHT | DataverseNL, 10.34894/5E8AHT |

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
