## [Editor Report · eLife Assessment]

This paper describes an **important** advance in a 2D in vitro neural culture system to generate mature, functional, diverse, and geometrically consistent cultures, in a 384-well format with defined dimensions and the absence of the necrotic core, which persists for up to 300 days. The well-based format and conserved geometry make it a promising tool for arrayed screening studies. The evidence is **compelling** and provides a method for generating consistent 3D cortical layer-like organization.

---

## [Referee Report · Reviewer #3 (Public review)]

Summary:

Kroeg et al. introduced a novel method for generating 3D cortical layer-like organization in hiPSC-derived models, achieving remarkably consistent topography within compact dimensions. Their approach involves seeding frontal cortex-patterned iPSC-derived neural progenitor cells into 384-well plates, which triggers the spontaneous assembly of adherent cortical organoids comprising diverse neuronal subtypes, astrocytes, and oligodendrocyte-lineage cells.

Strengths:

Compared with existing brain organoid models, these adherent cortical organoids exhibit enhanced reproducibility and improved cell viability during prolonged culture, thereby providing versatile opportunities for high-throughput drug discovery, neurotoxicological screening, and investigation of brain disorder pathophysiology. Overall, this study addresses an important and timely need for advancing current brain organoid systems.

Weaknesses:

Highlighting the consistency of differentiation across different cell lines and standardizing functional outputs are crucial to emphasize the broad future potential of this new organoid system for large-scale pharmacological screening. The authors provided a substantial amount of new data during the revision process to support the reproducibility of neuronal activity. The next step would be to leverage this platform for functional screening of chemical and genetic perturbations to identify new drug candidates.

Comments on revisions:

Most of my previous concerns were adequately addressed through the revision.

---

## [Author Response]

The following is the authors’ response to the original reviews.

**Public Reviews:**

**Reviewer #1 (Public review):**
Summary:Kroeg et al. describe a novel method for 2D culture human induced pluripotent stem cells (hiPSCs) to form cortical tissue in a multiwell format. The method claims to offer a significant advancement over existing developmental models. Their approach allows them to generate cultures with precise, reproducible dimensions and structure with a single rosette; consistent geometry; incorporating multiple neuronal and glial cell types (cellular diversity); avoiding the necrotic core (often seen in free-floating models due to limited nutrient and oxygen diffusion). The researchers demonstrate the method's capacity for long-term culture, exceeding ten months, and show the formation of mature dendritic spines and considerable neuronal activity. The method aims to tackle multiple key problems of in vitro neural cultures: reproducibility, diversity, topological consistency, and electrophysiological activity. The authors suggest their potential in high-throughput screening and neurotoxicological studies.Strengths:The main advances in the paper seem to be: The culture developed by the authors appears to have optimal conditions for neural differentiation, lineage diversification, and long-term culture beyond 300 days. These seem to me as a major strength of the paper and an important contribution to the field. The authors present solid evidence about the high cell type diversity present in their cultures. It is a major point and therefore it could be better compared to the state of the art. I commend the authors for using three different IPS lines, this is a very important part of their proof. The staining and imaging quality of the manuscript is of excellent quality.

We thank the reviewer for the positive comments on the potential of our novel platform to address key problems of in vitro neural culture, highlighting the longevity and reproducibility of the method across multiple cell lines.

Weaknesses:(1) The title is misleading: The presented cultures appear not to be organoids, but 2D neural cultures, with an insufficiently described intermediate EB stage. For nomenclature, see: doi: 10.1038/s41586-022-05219-6. Should the tissue develop considerable 3D depth, it would suffer from the same limited nutrient supply as 3D models - as the authors point out in their introduction.

We appreciate the opportunity to clarify this point. We respectfully disagree that the cultures do not meet the consensus definition of an organoid. In fact, a direct quote from the seminal nomenclature paper referenced by the reviewer states: “We define organoids as in vitro-generated cellular systems that emerge by self-organization, include multiple cell types, and exhibit some cytoarchitectural and functional features reminiscent of an organ or organ region. Organoids can be generated as 3D cultures or by a combination of 3D and 2D approaches (also known as 2.5D) that can develop and mature over long periods of time (months to years).” (Pasca et al, 2022 doi10.1038/s41586-022-05219-6). Therefore, while many organoid types indeed have a more spherical or globular 3D shape, the term organoid also applies to semi-3D or nonglobular adherent organoids, such as renal (Czerniecki et al 2018, doi.org/10.1016/j.stem.2018.04.022) and gastrointestinal organoids (Kakni et al 2022, doi.org/10.1016/j.tibtech.2022.01.006). Accordingly, the adherent cortical organoids described in the manuscript exhibit self-organization to single radial structures consisting of multiple cell layers in the z-axis, reaching ~200um thickness (therefore remaining within the limits for sufficient nutrient supply), with consistent cytoarchitectural topology and electrophysiological activity, and therefore meet the consensus definition of an organoid.

(2) The method therefore should be compared to state-of-the-art (well-based or not) 2D cultures, which seems to be somewhat overlooked in the paper, therefore making it hard to assess what the advance is that is presented by this work.

It was not our intention to benchmark this model quantitatively against other culture systems. Rather, we have attempted to characterize the opportunities and limitations of this approach, with a qualitative contrast to other culture methods. Compared to stateof-the-art 2D neural network cultures, adherent cortical organoids provide distinct advantages in:

(1) Higher order self-organized structure formation, including segregation of deeper and upper cortical layers.

(2) Longevity: adherent cortical organoids can be successfully kept in culture for at least 1 year, whereas 2D cultures typically deteriorate after 8-12 weeks.

(3) Maturity, including the formation of dendritic mushroom spines and robust electrophysiological activity.

(4) Cell type diversity including a more physiological ratio of inhibitory and excitatory neurons (10% GAD67+/NeuN+ neurons in adherent cortical organoids, vs 1% in 2D neural networks), and the emergence of oligodendrocyte lineage cells.

On the other hand, limitations of adherent cortical organoids compared to 2D neural network cultures include:

(1) Culture times for organoids are much longer than for 2D cultures and the method can therefore be more laborious and more expensive.

(2) Whole cell patch clamping is not easily feasible in adherent cortical organoids because of the restrictive geometry of 384-well plates.

(3) Reproducibility is prominently claimed throughout the manuscript. However, it is challenging to assess this claim based on the data presented, which mostly contain single frames of unquantified, high-resolution images. There are almost no systematic quantifications presented. The ones present (Figure S1D, Figure 4) show very large variability. However, the authors show sets of images across wells (Figure S1B, Figure S3) which hint that in some important aspects, the culture seems reproducible and robust.

We made considerable efforts to establish quantitative metrics to assess reproducibility. We applied a quantitative scoring system of single radial structures at different time points for multiple batches of all three lines as indicated in Figure S1C. This figure represents a comprehensive dataset in which each dot represents the average of a different batch of organoids containing 10-40 organoids per batch. To emphasize this, we have adapted the graph to better reflect the breadth of the dataset. Additional quantifications are given in Figure S2 for progenitor and layer markers for Line 1 and in Figure 2 for interneurons across all three lines, showing relatively low variability. That being said, we acknowledge the reviewer’s concerns and have modified the text to reduce the emphasis of this point, pending more extensive data addressing reproducibility across an even broader range of parameters.

(4) What is in the middle? All images show markers in cells present around the center. The center however seems to be a dense lump of cells based on DAPI staining. What is the identity of these cells? Do these cells persist throughout the protocol? Do they divide? Until when? Addressing this prominent cell population is currently lacking.

A more comprehensive characterization of the cells in the center remains a significant challenge due to the high cell density hindering antibody penetration. However, dyebased staining methods such as DAPI and the LIVE/DEAD panel confirm a predominance of intact nuclei with very minimal cell death. The limited available data suggest that a substantial proportion of the cells in the center are proliferative neural progenitors, indicated by immunolabeling for SOX2 (Figure 2A,D;Figure S4C). Furthermore, we are currently optimizing the conditions to perform single cell / nuclear RNA sequencing to further characterize the cellular composition of the organoids.

(5) This manuscript proposes a new method of 2D neural culture. However, the description and representation of the method are currently insufficient. (a) The results section would benefit from a clear and concise, but step-by-step overview of the protocol. The current description refers to an earlier paper and appears to skip over some key steps. This section would benefit from being completely rewritten. This is not a replacement for a clear methods section, but a section that allows readers to clearly interpret results presented later.

We have revised the manuscript to include a more detailed step-by-step overview of the protocol.

(b) Along the same lines, the graphical abstract should be much more detailed. It should contain the time frames and the media used at the different stages of the protocol, seeding numbers, etc.

As suggested, we have adapted the graphical abstract to include more detail.

**Reviewer #2 (Public review):**
Summary:In this manuscript, van der Kroeg et al have developed a method for creating 3D cortical organoids using iPSC-derived neural progenitor cells in 384-well plates, thus scaling down the neural organoids to adherent culture and a smaller format that is amenable to high throughput cultivation. These adherent cortical organoids, measuring 3 x 3 x 0.2 mm, self-organize over eight weeks and include multiple neuronal subtypes, astrocytes, and oligodendrocyte lineage cells.Strengths:(1) The organoids can be cultured for up to 10 months, exhibiting mature dendritic spines, axonal myelination, and robust neuronal activity.(2) Unlike free-floating organoids, these do not develop necrotic cores, making them ideal for high-throughput drug discovery, neurotoxicological screening, and brain disorder studies.(3) The method addresses the technical challenge of achieving higher-order neural complexity with reduced heterogeneity and the issue of necrosis in larger organoids. The method presents a technical advance in organoid culture.(4) The method has been demonstrated with multiple cell lines which is a strength.(5) The manuscript provides high-quality immunostaining for multiple markers.

We appreciate the reviewer’s acknowledgement of the strengths of this novel platform as a technical advance in organoid culture that reduces heterogeneity and shows potential for higher throughput experiments.

Weaknesses:(1) Direct head-to-head comparison with standard organoid culture seems to be missing and may be valuable for benchmarking, ie what can be done with the new method that cannot be done with standard culture and vice versa, ie what are the aspects in which new method could be inferior to the standard.

In our opinion, it would be extremely difficult to directly compare methods. Most notably, whole brain organoids grow to large and irregular globular shapes, while adherent cortical organoids have a more standardized shape confined by the geometry of a 384well. Moreover, it was not our intention to benchmark this model quantitatively against other culture systems. Rather, we have attempted to characterize the opportunities and limitations of this approach, with a qualitative contrast to other culture methods, as addressed in response to comment 2 of Reviewer 1 above.

(2) It would be important to further benchmark the throughput, ie what is the success rate in filling and successfully growing the organoids in the entire 384 well plate?

Figure S1 shows the success rate of organoid formation and stability of the organoid structures over time. In addition, we have added the number of wells that were filled per plate.

(3) For each NPC line an optimal seeding density was estimated based on the proliferation rate of that NPC line and via visual observation after 6 weeks of culture. It would be important to delineate this protocol in more robust terms, in order to enable reproducibility with different cell lines and amongst the labs.

Figure S1 provides the relationship between proliferation rate and seeding density, allowing estimation of seeding densities based on the proliferation rate of the NPCs. However, we appreciate the reviewers' feedback and have modified the methods to provide more detail.

**Reviewer #3 (Public review):**
Summary:Kroeg et al. have introduced a novel method to produce 3D cortical layer formation in hiPSC-derived models, revealing a remarkably consistent topography within compact dimensions. This technique involves seeding frontal cortex-patterned iPSC-derived neural progenitor cells in 384-well plates, triggering the spontaneous assembly of adherent cortical organoids consisting of various neuronal subtypes, astrocytes, and oligodendrocyte lineage cells.Strengths:Compared to existing brain organoid models, these adherent cortical organoids demonstrate enhanced reproducibility and cell viability during prolonged culture, thereby providing versatile opportunities for high-throughput drug discovery, neurotoxicological screening, and the investigation of brain disorder pathophysiology. This is an important and timely issue that needs to be addressed to improve the current brain organoid systems.

We thank the reviewer for highlighting the strengths of our novel platform. We appreciate that all three reviewers agree that the adherent cortical organoids presented in this manuscript reliably demonstrate increased reproducibility and longevity. They also commend its potential for higher throughput drug discovery and neurotoxicological/phenotype screening purposes.

Weaknesses:While the authors have provided significant data supporting this claim, several aspects necessitate further characterization and clarification. Mainly, highlighting the consistency of differentiation across different cell lines and standardizing functional outputs are crucial elements to emphasize the future broad potential of this new organoid system for large-scale pharmacological screening.

We appreciate the feedback and have added more detail on consistency and standardization of functional outputs.

**Recommendations for the authors:**

**Reviewer #1 (Recommendations for the authors):**
Minor points(1) As the preprint is officially part of the eLife review, I have to remark that the preprint which is made available on bioarxiv, suffers from some serious compatibility or format problem: one cannot highlight sentences as in a regular PDF and when trying to copypaste sentences from it jumbled characters are copied to the clipboard.

The updated version of the paper on *bioRxiv* should not suffer from these compatibility issues.

(2) Since the paper is presenting a new method it should briefly describe how each step, including the hiPSC culture was done, the reference to an earlier publication in this case is not sufficient, and this practice is generally best to avoid for methods papers.

Each step in the culturing process has now been described in the methods.

(3) The EB stage is insufficiently described. The "2D - 3D - 2D" transitions should be clearly explained.

The methods section has been rewritten and expanded to include these processes in more detail.

(4) Is there one FACS sorting in the protocol, or multiple (additional at IPS culture)? What markers each? What is the motivation for sorting and purifying the neural progenitors? Was the culture impure? What was purity? What cell types are expected after sorting, and what is removed?

Only one FACS sorting step is performed at the NPC stage. This was added as an improvement to our original neural network protocol (Günhanlar et al 2018) to ensure consistency over different hiPSC source cell lines that can yield variable amounts of frontal cortical patterned NPCs. Positive sorting for neural lineage markers CD184 and CD24, and negative sorting for mesenchymal/neural crest CD217 and CD44 glial progenitor markers, according to Yuan et al 2011, ensures frontal-patterned cortical NPCs as confirmed for all batches by immunohistochemistry for SOX2, Nestin and FOXG1. We have added new text to the Methods section to clarify this more explicitly.

(5) Seeding protocol and parameters are insufficiently described, and from what I read they are poorly defined: "Specifically, the optimal seeding density was determined by visual inspection of the organoids between 28 to 42 days after seeding a range of cell densities in the 384-well plate wells." For a new method, precise, actionable instructions are needed. I may have overlooked those elsewhere, in this case, please clarify these sections.

The Methods section was rewritten and expanded to describe the methodology in greater detail with more actionable instructions.

(6) The timeline in Figure 1 is not clearly delineated; I found it hard to understand which figure corresponds to which stage (e.g. facs sorting is not mentioned in the first part of the results but it is part of Figure 1A, neural rosette formation can happen both before and after facs sorting, simply referring to rosettes is not clear). Later parts of the manuscript > clearly introduce the terms sorting and seeding in the context of this method, and how ages (days) refer to these time points.

Figure 1 was adapted to clarify the generation of Neural Progenitor Cells (NPCs) and subsequent seeding of NPCs to generate Adherent Cortical Organoids (ACOs).

(7) The authors define: "cortical organized defined as a single radial structure." This is not a commonly used definition of organoids, for nomenclature, please see: doi: 10.1038/s41586-022-05219-6 (Pasca et al 2022).

To clarify, the statement is not meant to reflect a definition of organoids in general, but rather the scoring of proper structure formation for Figure S1C. For discussion on nomenclature, see our response to point 1 of Reviewer 1 in the public review. We changed the wording to be more accurate.

(8) In Figure S1d, the authors write: "the fraction of structurally intact cultures decreased to 50%", but I'm looking at that graph there seems to be no notable decrease, but huge variability. The authors should quantify claims of decrease by linear regression and an R square. Variation within and the cross-cell lines seem to be large. Also, it is unclear if dots are corresponding to the same wells/plates, in other words: is this a longitudinal experiment? What is the overall success rate? How is success determined? Are there clear criteria? to the same wells/plates, in other words: is this a longitudinal experiment? What is the overall success rate? How is success determined? Are there clear criteria?

We agree with the reviewer that the claim on fraction of intact cultures decreasing over time to 50% is an overinterpretation due the large variability. We changed the wording in the manuscript to: While some later batches show moderately reduced success rates compared with the earliest batches, properly formed single-structure organoids were still obtained at 40–90% success across all examined time points (Figure S1C), indicating that long-term culture is feasible albeit with variable efficiency. The data are not longitudinal as each dot represents an endpoint of a different batch of organoids, totaling 18 independent batches across the three lines. We have clarified this in the figure legend. Success was defined at the well level as the presence of a single, continuous radial structure occupying the well, without obvious fragmentation or fusion events, as assessed by LIVE/DEAD that also confirmed viability. Wells were scored as successful only when the radial structure showed predominantly live signal with no large necrotic areas. Wells containing multiple radial structures, fused aggregates, or predominantly dead tissue were scored as unsuccessful.

(9) Figure s1c: the numbering to this panel should be swapped, because it is referenced after other panels in the text. The reference is confusing: "Plotting the interaction between proliferation and the amount of NPCs required to be seeded for the successful generation of adherent cortical organoids" - success is not present in this graph at all? How is that measured?

Figures S1C and S1D have been adapted to clarify the measure of ‘successful organoid formation’.

(a) The description of this plot is confusing: "The doubling time of the NPCs explains more than half the variation (r2 = 0.67) of the required seeding density." What else is there? I thought that this was the formula the authors suggested to determine seeding density, but it seems not. Or is "manual inspection" the determinant, and that seems to correlate with this metric?

Even though the rate of proliferation, measured as doubling time, is the main determinant of the seeding density, it is not the only determinant of the seeding density. For instance, intrinsic differences in differentiation potential could also play a role. Therefore, NPC lines with similar doubling times might still have slightly different optimal seeding densities. We have added clarification of this conclusion to the Results section.

(b) Seeding density is a key parameter in many in vitro differentiation and culture protocols. This importance however does not mean that this density is attributable to differences in cell proliferation rate. Alternatively, the amount of cells determines the amount of secreted molecules and cell-to-cell contacts.

Here, when we refer to the cell density, we specifically refer to the cell density needed to generate the ACO. We show that the most important contributor to the variation in ACO formation is the proliferation, measured here as the doubling time. We agree that there are other factors involved such as the secreted molecules, cell-to-cell contacts as well as the ability of a given NPC line to differentiate into a post-mitotic cell.

(c) Is it mentioned which cell line this experiment corresponds to?

The data in Figure S1D is from the 3 reported cell lines, as well as 2 clones from a fourth IPS cell line. This is detailed in the Methods section of the proliferation assay.

(d) Without a more detailed explanation, seeding density and doubling time could be independent variables.

These two variables are highly correlated as shown in Figure S1D, but it is true that there can be other variables that account for the observed variance, as discussed above in Point 9b.

(e) In this figure the success rate is not visible at all so I have no idea how the autors arrive at a conclusion about success rate.

We have adapted the figure legend to reflect which cell lines the dots in Fig. S1D represent. NPC lines can have substantial variation in proliferation rates. The figure reflects data of NPCs of 5 clones of 4 different hiPSC lines (as indicated in the Methods) with different proliferation rates. Also, the ACO success rate (operationally defined uniformly to the data shown in Fig. S1C) was also included.

(10) Figure 2: Clean spatial segregation seems to be a strength of the system and therefore I would recommend putting more of the relevant microscopy images to the main figure, which are now currently in Figure S4.

We have adapted Figure 2 accordingly, and included additional representative cortical layering images in Figure S4.

(11) The variability in interneuron content seems to be significant, as currently presented in the figure. However, this may be due to a special organization. It would first quantify in consecutive rings around the centers whether interneurons have a tendency to be enriched towards the center or the edge of the culture. Maybe this explains the variability that is currently present in Figure s5b.

We agree that spatial organization of interneurons could, in principle, contribute to variability. In our analysis, however, images were acquired from positions selected by a random sampling grid across the entire culture, rather than from specific central or peripheral regions. Each field contained on average 130.6 ± 16.1 NeuN+ nuclei, which provided a relatively large sampling volume per position. If interneurons were strongly enriched at the center or edge, we would expect systematic differences in interneuron fraction between fields assigned to central versus peripheral grid positions. We did not observe such a pattern in our dataset, suggesting that spatial organization is not the main driver of the observed variability.

(12) Because in previous figures it seems like there is considerable variability across individual cultures and images here are coming from separate cultures, please use different shapes of the points coming from different cultures/wells, to see if maybe there is a culture-to-culture difference that explains the variability present in the figure.

We have added different symbols per organoid for the interneuron quantifications and moved this quantification to main Figure 2.

(13) I believe it is currently the standard error of the mean which is displayed in the figure, which is not an appropriate representation for variability, or the reproducibility across individual data points. SEM quantifies the reproducibility of the mean, not the reproducibility of the individual data points, which matters here. Mean refers to the mean of this quantification experiment and therefore it's not a biological entity. A box plot showing the interquartile range besides the individual data points would be an accurate representation of the spread of the data.

We agree and have adapted the data, now in Figure 5, accordingly.

(14) Again, in general, the main figures should contain much more of the quantification, as opposed to just raw images.

Quantifications have been added in Figure 2 for the GAD67/NeuN for all cell lines as well as a time course quantification of GAD67/NeuN for 1 of the cell lines. In Figure 4, we have added excitatory and inhibitory synaptic quantifications.

(15) Figure 2F-I the location of the center of the rosette should be marked with a star so that the conclusion about the direction of processes can be established.

The suggested addition of a marker at the center of each rosette was evaluated but not implemented, because it reduced rather than improved figure clarity.

(16) Figure 3 b and c:High magnification images of single cells, can't show changes in cell type morphology, and one cannot conclude that these cells are present in significant numbers across time. Zoomed-out images or quantification would be necessary for such a claim. The authors already have such images as presented in the next panels, so quantification without new experiments. > I am uncertain about the T3 supplement here - do these images correspond to the same conditions?(a) It is unclear to me why different markers are used in the different panels, namely why NG2 is not used in any of the other images.

NG2 was used at early developmental time points to show the presence of Oligodendrocyte Precursor Cells (OPCs). At later time points, the focus switched to MBP staining to indicate more mature oligodendrocyte lineage cells. Although NG2 and MBP are not in the same panels, the staining was performed for both antibodies at the same developmental time point (Day 119) as seen in Figure 3C and 3D.

(b) Color coding in Figure 3G is ambiguous; the use of two blues should be avoided, and the Sub-sub panels should be individually labeled for the color code.

We agree, and have now used different colors.

(c) It is unclear if the presence of the t3 molecule is part of the standard procedure or if it was a side experiment to enhance the survival of oligodendrocytes. Are there no oligodendrocytes without? How does T3 affect other cell types, and the general health and differentiation of the cultures?

Indeed, T3 is essential for oligodendrocyte formation. We did not observe obvious effects on the general health or differentiation potential of the cultures.

(d) Is the 2ng/ml t3 from day one to the final day?

Indeed, in the organoids cultured to study oligodendrocyte formation, T3 was added from Day 1. These details have now been clarified in the Methods and Results sections.

(17) Figure 4:(a) Microscopy in this figure is high quality and very convincing about neural maturity.(b) The term "cluster" should be avoided. Unclear what it means here, but my best guess is "cells in a frame of view." Cluster is used with a different meaning in electrophysiology.

This was adapted to ‘neurons in a field of view (FOV)’.

(c) Panel J: I assume each row corresponds to a single cell? Could this be clarified? Are these selected cells from each frame, or all active cells are represented?

Indeed, each row corresponds to a single cell, showing all active cells in the frame. This is now clarified in the legend.

(d) How many Wells do these data correspond to, and in which line it was measured?

As reported in the legend for Figure 5, these data correspond to 2 wells at Day 61 to which we have now added calcium imaging data from 3 wells from a different batch at Day 100. We have included in the legend that these recordings were from Line 1.

(e) Panels G to I, again, the use of standard error of the mean is inappropriate and misleading: looking at the error bar one must conclude that there is minimal variation, which is the exact opposite of the conclusions, when one would look at the variability of the raw data points.

As suggested, the graphs have been adapted as boxplots with interquartile ranges to highlight the distribution of data points.

(f) It is unclear how many neurons and how many total actively firing neurons are present in the videos analyzed

All neurons that were active in the field of view and showed at least one calcium event during the ~10 minute recording were included in the analysis. Using this method, we cannot comment on the proportion of neurons that were active from the total amount of neurons present, since the AAV virus we used does not transduce all neurons.

(g) This figure shows the strength of the method in achieving neural maturity and function. There seems to be that there is considerable activity in the neuronal cultures analyzed. To conclude how reliably the method leads to such mature cultures one would need to measure at least a dozen wells (even if with some simpler and low-resolution method). Concluding reproducibility from one or two hand-picked examples is not possible.

We agree with the reviewer that the number of wells used for calcium imaging analysis was limited. We are currently working on more advanced methods to increase the throughput of this analysis. However, we’ve now added another timepoint to the calcium imaging data in Figure 5 from an independent batch of 3 adherent cortical organoids, which demonstrates continued robust activity at Day 100, as well as Day 61.

Methods:(1) Stem cell culture. The artist described that line 3 is grown on MEFs. Is this true for the other two lines, furthermore were they cultured in identical conditions?

Line 2 and 3 were not grown on MEFs. We specifically chose different sources of NPCs to reflect the robust nature of the differentiation protocol. We have recently also adapted the protocol from Line 3 NPCs to confirm that the protocol also works starting from hiPSCs grown in feeder-free conditions in StemFlex medium, by adapting NPC differentiation according to our recent publication in *Frontiers in Cellular Neuroscience* (Eigenhuis et al 2023).

(2) "NPCs were differentiated to adherent cortical organoids between passages 3 and 7 after sorting." Please clarify this sentence. I assume it refers to the first facs sorting of the protocol, but a section is not sufficiently detailed.

We have adapted the methods to clarify that the FACS purification step occurs at the NPC stage.

(3) I didn't fully understand: It seems to be that there are two steps of fact sorting involved, one after passage 3 and one after week 4. This should be represented in the graphical abstract of Figure 1.

As outlined above, there is only 1 FACS sorting step at NPC stage. We have adapted this in the Methods and in the graphical abstract.

(4) Neural differentiation: The authors write that optimal seeding density was determined by visual inspection of the organoids - this is.

We have clarified the Methods section to better explain the process of optimizing the seeding density for each NPC line to generate the ACOs.

(5) What does the following sentence mean: "Cells were refreshed every 2-3 days." Does it mean in replacement of the complete media? How much Media was added to the Wells?

This is a very good point that we have now clarified in the Methods, as full replenishment of media is neither feasible, nor desirable. From the total volume of 110 µl per well, 80 µl is taken out and replaced with 85 µl to compensate for evaporation.

(6) Calcium imaging: can the authors explain the decision to move the cultures one day before imaging into brainphys neural differentiation medium? In 3D organoid protocols, brainphys is gradually introduced to avoid culture shock (very different composition), and used for multiple months to enhance neural differentiation. For recording electrophysiological activity, artificial CSF is the most common choice.

Indeed, for whole cell recordings of 2D neural networks as performed in Günhanlar et al 2018, we used gradual transition to aCSF. For the current ACOs, we found that using BrainPhys from the start of organoid differentiation prevents structure formation, probably because of increased speed of maturation disrupting proliferation and organization of radial glia differentiation. However, by changing the media to BrainPhys just one day before recording (reflecting a gradual change as not all medium is fully replenished and easier than switching to aCSF during recording), we saw greatly improved neuronal activity.

(7) Statistical analysis : As I pointed out before, the standard error of the mean is not an appropriate metric to represent the variability of the data. It is meant to represent the variability of the estimated average. The following thought experiment should make it clear: I measured the expression of a gene in my system. 50 times I measured 0 and 50 times I measured 100. The average is 50, but of course it is a very bad representation of the data because no such data points exist with that value. Yet the standard error of the mean would be plus minus 5.

We have revised Figures 5C–5D to boxplots displaying the interquartile range with all individual data points overlaid, which more accurately represents the variability in the dataset.

Discussion(1) The discussion focuses on human cortical development, however, the methods presented by the authors entail dissociation and replating through multiple stages not part of brain development. I see the approach as more valuable as a possibly reliable method that generates both diverse and mature neural cultures.

We have revised the Discussion to avoid explicitly invoking an in vitro recapitulation of human cortical development. Nevertheless, given that the NPCs from which the organoids originate exhibit frontal cortical identity, coupled with the timely emergence of cortical neuronal markers and rudimentary cortical layering, we are increasingly confident that the development of these cultures most likely mirrors that of the frontal cortex. To further substantiate this hypothesis, single-cell RNA sequencing experiments will be conducted in the future to provide additional insights.

(2) One of the major claims of the authors is that the method is very reproducible. However, there is almost no data on reproducibility throughout the paper. Mostly single, high magnification images are presented, which therefore represent a small region of a single well of a single batch of a single cell line. Based on the data presented it is not possible to evaluate the reproducibility of the method.

We agree that the original version did not sufficiently document reproducibility. To address this, we have refined and expanded our presentation of reproducibility data. The previous success-rate panel (original Figure S1D) has been moved and adapted as the new Figure S1C. In this updated version, each dot still represents the endpoint success rate of an independent batch, but dot size now scales with batch size (10–40 organoids), and the legend specifies the total numbers of organoids analyzed per line (line 1: n=248; line 2: n=70; line 3: n=70). Together with the distribution of success rates between ~40– 90% across multiple time points and three iPSC lines, this more detailed representation allows readers to directly assess the robustness of line-to-line and batch-to-batch performance. In addition, new time course quantifications of interneuron proportion (Figure 2G,H), synaptic marker densities (Figure 4H, I), and late-stage calcium imaging (Figure 5C,D,E) further demonstrate that key structural and functional read-outs show overlapping ranges across lines and independent differentiations, reinforcing that the method yields reproducible core phenotypes despite some biological variability.

(3) The data presented is very promising, and it suggests that the authors derived optimal conditions for neural differentiation and neural culture diversification. I am confident that the authors can show that reproducibility, at least in a practical sense (e.g. in wells that form a culture) is high.Overall, this is a very promising and exciting work, that I am looking forward to reading in a mature manuscript.
**Reviewer #2 (Recommendations for the authors):**
(1) Direct head-to-head comparison with standard organoid culture seems to be missing and may be valuable for benchmarking, ie what can be done with the new method that cannot be done with standard culture and vice versa, ie what are the aspects in which new method could be inferior to the standard.

We have now more clearly elaborated the differences with other methods. As addressed in our response to point 2 of Reviewer 1 in the public reviews, there are several limitations and advantages to the adherent cortical organoids model listed as follows:

Advantages of adherent cortical organoids:

(1) Higher order self-organized structure formation, including segregation of deeper and upper cortical layers.

(2) Longevity: adherent cortical organoids can be successfully kept in culture for at least 1 year, whereas 2D cultures typically deteriorate after 8-12 weeks.

(3) Maturity, including the formation of dendritic mushroom spines and robust electrophysiological activity.

(4) Cell type diversity including a more physiological ratio of inhibitory and excitatory neurons (10% GAD67+/NeuN+ neurons in adherent cortical organoids, vs 1% in 2D neural networks), and the emergence of oligodendrocyte lineage cells.

On the other hand, limitations of adherent cortical organoids compared to 2D neural network cultures include:

(1) Culture times for organoids are much longer than for 2D cultures and the method can therefore be more laborious and more expensive.

(2) Whole cell patch clamping is not easily feasible in adherent cortical organoids because of the restrictive geometry of 384-well plates.

(2) It would be important to further benchmark the throughput, ie what is the success rate in filling and successfully growing the organoids in the entire 384 well plate?

We have addressed this question in the current version of Fig. S1C, in which multiple batches of organoids of all three lines were scored for their success rate. The graph reflects the proportion of properly formed organoids of +/- 400 seeded wells scored at different timepoints, in which each timepoint is a different batch. As mentioned in the response to Reviewer 1, we have also added data on the number of organoids seeded per line in the figure legend.

(3) For each NPC line an optimal seeding density was estimated based on the proliferation rate of that NPC line and via visual observation after 6 weeks of culture. It would be important to delineate this protocol in more robust terms, in order to enable reproducibility with different cell lines and amongst the labs.

As outlined in the response to Reviewer 1, we have clarified the Methods and Discussion sections on seeding density and proliferation rate.

**Reviewer #3 (Recommendations for the authors):**
Kroeg et al. have introduced a novel method to produce 3D cortical layer formation in hiPSC-derived models, revealing a remarkably consistent topography within compact dimensions. This technique involves seeding frontal cortex-patterned iPSC-derived neural progenitor cells in 384-well plates, triggering the spontaneous assembly of adherent cortical organoids consisting of various neuronal subtypes, astrocytes, and oligodendrocyte lineage cells. Compared to existing brain organoid models, these adherent cortical organoids demonstrate enhanced reproducibility and cell viability during prolonged culture, thereby providing versatile opportunities for high-throughput drug discovery, neurotoxicological screening, and the investigation of brain disorder pathophysiology. This is an important and timely issue that needs to be addressed to improve the current brain organoid systems. While the authors have provided significant data supporting this claim, several aspects necessitate further characterization and clarification. Particularly, highlighting the consistency of differentiation across different cell lines and standardizing functional outputs are crucial elements to emphasize the future broad potential of this new organoid system for large-scale pharmacological screening.(1) Considering the emergence of astrocyte markers (GFAP, S100b) and upper layer neuron marker (CUX1) around Day 60, the overall differentiation speed is significantly faster compared to other forebrain organoid protocols. Are these accelerated sequences of neurodevelopment consistent across different hiPSC lines?

As shown in Fig. S5, astrocytes are present around Day 60 for all three lines. For comparison with other organoid protocols, an important consideration is that the timeline for these organoids starts at NPC plating, while for other protocols timing often starts from the hiPSC stage. We have clarified the timeline in the graphical abstract in Figure 1A and in the Methods.

(2) The calcium imaging results in Figure 4G were recorded at a single time point, Day 61, a relatively early time window compared to other forebrain organoid protocols (more than 100 days, PMID: 31257131; PMID: 36120104). Are the neurons in adherent cortical organoids functionally mature enough around Day 61? How consistent is this functional activity across different cell lines and independent differentiation batches?

As discussed above in Point 1, it is important to consider that the specified timeline starts from NPC plating. In analogy to 2D neural networks, robust neuronal activity can be observed after ~8 weeks in culture. In addition, we have now added calcium imaging data for an additional batch of organoids at Day 100 in Figure 5, which exhibit comparable levels of neuronal activity as observed on Day 61.

(3) Along the same line, Various cell types, such as oligodendrocytes and astrocytes, are believed to influence neuronal maturation. Therefore, longitudinal studies until the late stage are necessary to observe changes in electrophysiological activity based on the degree of neuronal maturation (at least two more later time points, such as 100 days and 150 days).

As described in the previous points, we have now included a Day 100 time point in the calcium imaging data, in addition to the recordings at Day 61 (Figure 5C-E).

(4) The authors assert that heterogeneity among organoids has been diminished using the human adherent cortical organoids protocol. However, there is inadequate quantitative data to prove the consistency of neuronal activities between different wells. Therefore, experiments quantifying the degree of heterogeneity between organoids, such as through methods like calcium imaging, are necessary to determine if neuron activity occurs consistently across each organoid well.

We agree with the review and have added several quantitative experiments: (a) we’ve added another timepoint to the calcium imaging data in Figure 5 from an independent batch of 3 adherent cortical organoids, which demonstrates continued robust activity at day 100, as well as day 61; (b) we added synapse quantification in Figure 4, and (c) interneuron quantification in Figure 2. We are currently also pursuing high throughput measures of activity to assess the longitudinal activity of ACOs in a larger number of wells. This way we can more definitively quantify the time-dependent variance in organoid activity.

(5) Is this platform applicable to other functional measurements for neuronal activity, such as the MEA system? When observing the morphology of neurons formed in organoids, they appear to extend axons and dendrites in a consistent direction, suggesting a radial structure that demonstrates high reproducibility across wells. A culture system where neurons are arranged with such consistency in directionality could be highly beneficial for experiments utilizing the MEA system to assess parameters such as the speed of electrical activity transmission and stimulus-response. Therefore, there seems to be a need for a more detailed explanation of the utility of the structural characteristics of the culture system.

The ACO platform is indeed suitable for MEA recordings. We are in the process of engineering the required geometry using HD-MEA systems through specialized inserts to generate ACOs on MEA systems.

(6) In Figure 2E-I, authors suggest morphological diversity of GFAP+/S100b+ astrocyte, but the imaging data presented in Figure F-I is only based on GFAP immunoreactivity.Since GFAP is also expressed in radial glial cells at this stage (Figure 2I), many fibrous astrocytes and interlaminar astrocytes are likely radial glial neural progenitor cells instead of astrocytes. It appears necessary to perform additional staining using astrocyte markers such as S100B or outer radial glia markers such as HOPX to demonstrate that the figure depicts subtype-specific morphologies of astrocytes.

In Figure 2M, we stained for GFAP and PAX6 to mark radial glia that look different than the astrocyte morphologies we describe in Figure 2J-L. We see a large overlap in GFAP and S100B staining in Figure 2I, in which most GFAP+ cells are double positive for S100B (yellow) that is more consistent with astrocyte maturation than radial glia. Furthermore, we have not seen PAX6 staining outside the dense edges of the center of the ACO.

(7) In Figure 4D, the axon appears to exhibit directionality. Additional explanation regarding the organization of the axon is necessary. Further research utilizing sparse staining to examine the morphology of single neurons seems warranted.

The polarized directionality of the axons is something we indeed have also noticed. We are looking into options to further investigate this intriguing property of the ACOs.

(8) Figure 1E-F only showed cell viability in the early stages around Day 40-50. To demonstrate the superior long-term viability of ACO culture, it appears necessary to illustrate the ratio of dead cells to live cells over the course of a time course.

Figure S1B shows LIVE/DEAD staining for ACOs of all three lines, revealing minimal DEAD staining at Day 56. A longitudinal time course experiment was not performed, however the line- and batch-specific quantifications over developmental timepoints in Figure S1C provide an indication of the robust long-term viability of the ACOs.